# The *Aphelenchus avenae* genome highlights evolutionary adaptation to desiccation

Xuehua Wan [1,2✉], Jennifer A. Saito[1], Shaobin Hou[1], Scott M. Geib[3], Anton Yuryev[4], Lynne M. Higa[5], Christopher Z. Womersley [5✉] & Maqsudul Alam[1,6]

Some organisms can withstand complete body water loss (losing up to 99% of body water) and stay in ametabolic state for decades until rehydration, which is known as anhydrobiosis. Few multicellular eukaryotes on their adult stage can withstand life without water. We still have an incomplete understanding of the mechanism for metazoan survival of anhydrobiosis. Here we report the 255-Mb genome of *Aphelenchus avenae*, which can endure relative zero humidity for years. Gene duplications arose genome-wide and contributed to the expansion and diversification of 763 kinases, which represents the second largest metazoan kinome to date. Transcriptome analyses of ametabolic state of *A. avenae* indicate the elevation of ATP level for global recycling of macromolecules and enhancement of autophagy in the early stage of anhydrobiosis. We catalogue 74 species-specific intrinsically disordered proteins, which may facilitate *A. avenae* to survive through desiccation stress. Our findings refine a molecular basis evolving for survival in extreme water loss and open the way for discovering new anti-desiccation strategies.

[1] Advanced Studies in Genomics, Proteomics and Bioinformatics, University of Hawaii, Honolulu, HI, USA. [2] TEDA Institute of Biological Sciences and Biotechnology, Nankai University, Tianjin, P. R. China. [3] Tropical Crop and Commodity Protection Research Unit, USDA-ARS Pacific Basin Agricultural Research Center, Hilo, HI, USA. [4] Elsevier Life Sciences Solutions, Rockville, MD, USA. [5] School of Life Sciences, University of Hawaii, Honolulu, HI, USA. [6] Deceased: Maqsudul Alam. ✉email: xuehua.wan@hotmail.com; womersle@hawaii.edu

Although water is essential for life, certain organisms, across the three kingdoms of life, can survive extreme water loss by entering ametabolic state, known as anhydrobiosis, in the certain stage of life cycle (e.g., metazoan larvae or/and adult stage)[1]. Few metazoan, such as rotifers, tardigrades, and nematodes, can endure complete water loss in their adult stage. Genomes were decoded for bdelloid rotifers and tardigrades[2–4], providing rotifer- and tardigrade-unique proteins as protective molecules. Massive horizontal gene transfer (HGT) events were identified in bdelloid rotifer *Adineta vaga* genome[2,5], whereas no extensive HGT was found in tardigrade *Ramazzottius varieornatus* and *Hypsibius dujardini* genomes[3,4,6]. Progresses were made to understand the molecular mechanisms for these organisms to enter anhydrobiosis[7–10]. However, important species-specific protective molecules and antidesiccation mechanisms in nematode species remain to be determined.

Nematoda, a species rich Phylum, distributes in diverse habitats, such as terrestrial and aquatic ecosystems as well as animal and human hosts. *A. avenae*, certain other species in the suborder Tylenchina, and Antarctic nematodes such as *Plectus murrayi* from the class Chromadorea undergo anhydrobiosis, whereas species in other suborders can only tolerate desiccation in dauer juvenile stage or larvae within egg[11–16]. The genome and transcriptome of *P. murrayi*, which inhabits the Antarctic ecosystem, have recently been characterized[17]. As an anhydrobiotic model, *A. avenae* has derived the dormancy capability to escape cell death in relative zero humidity for years[18]. When water is available, anhydrobiotic *A. avenae* resumes normal metabolic activity, suggesting that *A. avenae* has developed molecular shield systems to protect somatic cells during long-term dehydration[18]. *A. avenae* can only bear a slow graduate rate of water loss and is unable to survive without preconditioning at 97% relative humidity[19]. Nonreducing trehalose and late embryogenesis abundant (LEA) proteins were proposed to act synergistically to form gel-phase bioglass in anhydrobiotic *A. avenae*[18,20]. It is not completely understood how the genetic changes in the evolution of anhydrobiotic nematodes enable resistance to complete desiccation and how *A. avenae* reprograms its cells state in different dehydration stages for survival in complete water loss.

Here we report a high-quality genome of *A. avenae* along with its transcriptomal changes in response to gradual water loss. Comparative genomic and transcriptomic analyses will help understand general and species-specific molecular mechanisms for complex animal to survive in extreme water loss.

## Results and discussion

**Genome features of *A. avenae***. We estimated the genome size of *A. avenae* to be 255 Mb based on flow cytometry analysis, which is 2.5 times the size of *Caenorhabditis elegans*. To decode anhydrobiotic genes, we sequenced the genome of *A. avenae* at ~180-fold coverage using Roche/454 and Illumina platforms (Supplementary Table 1). K-mer counting further supports the 255 Mb genome size (Supplementary Fig. 1). The draft assembly consists of 18,660 scaffolds totaling 264.8 Mb, with an N50 of 142 kb ($n = 385$) and maximum scaffold length of 5.5 Mb (Fig. 1a, Supplementary Table 2, Supplementary Note). Total 218.6 Mb bases, representing 98% of the assembled genome sequences, had more than 20 reads covered (Supplementary Fig. 2), showing high single-base accuracy. Repetitive sequences covered 16.7% of the genome, most of which were uncharacterized repeats (Supplementary Table 3). The average density of single nucleotide polymorphisms (SNP) was about three variants per kb (Fig. 1a, Supplementary Note), confirming that *A. avenae* individuals in a population are highly identical due to parthenogenesis. We determined the assembly to be at least 97% complete, based on

the mapping of conserved eukaryotic clusters of orthologous groups (KOGs) (Supplementary Note). Evaluation of genome completeness based on BUSCO identified 236 (92.6%) of the 255 conserved BUSCO genes[21] (Supplementary Note). Moreover, all of the 5120 expressed sequence tags (EST) retrieved from NCBI can be mapped to the assembled scaffolds, indicating the completeness of the assembly.

We predicted 43,192 protein-coding genes, which occupied 15% of the genome at an average gene density of 169 genes per Mb (Supplementary Table 2). The assembled transcriptome supported 80% (34,387) of the gene models. Compared to 17% of *C. elegans* protein-coding genes organized into operons[22], 23.3% of *A. avenae* protein-coding genes were predicted in operons (Fig. 1a, Supplementary Note, Supplementary Data 2). We assigned functions to 21,910 (50.7%) protein-coding genes. BLASTP identified 15,645 protein sequences (36.2%) shared in the phylum Nematoda and 19,080 sequences (44.2%) as species-specific orphan genes (Supplementary Note).

**Evolution of *A. avenae* genome**. To describe gene family emergence and extinction that underline species adaptation, we compared eight nematode genomes including *A. avenae*, *C. elegans*, *B. malayi* (causes lymphatic filariasis), *Trichinella spiralis* (causes trichinellosis), *Meloidogyne hapla* (causes root-knot), *Ascaris suum* (causes ascariasis), *Bursaphelenchus xylophilus* (causes pine wilt), and *Haemonchus contortus* (causes haemonchosis). Only 998 gene families were conserved across the eight nematode species, implying that births and deaths of gene families diversely occurred in the phylum Nematoda. Gene birth/death tree showed that *A. avenae* and *B. xylophilus* shared the highest number of gene families compared to the other six nematode species (Fig. 1b). A total of 26,537 (61.4%) proteins were unique to *A. avenae*, which play roles in immune system, developmental process, and response to stimulus (Supplementary Fig. 3). Additional comparison of *A. avenae* and clade IV species showed that 20,615 (47.7%) proteins were unique to *A. avenae* (Supplementary Note, Supplementary Table 7).

Next, we examined the genome rearrangements between *A. avenae* and the selected nematode species to reveal synteny relationships that evolved for million years. Scaffolds/chromosomes above 100 kb were pairwisely compared. Analyzing whole-genome alignments and clustering highly conserved linkages, we found that synteny between *A. avenae-B. xylophilus* (3.4%) was dramatically higher than those either between *A. avenae-C. elegans* (0.5%) or *A. avenae-M. hapla* (0.5%) (Supplementary Fig. 4). *A. avenae* and *B. xylophilus* shared sixty syntenic blocks, while *A. avenae* and *C. elegans* shared eleven syntenic blocks, suggesting *A. avenae* and *B. xylophilus* diverged later than *A. avenae* and *C. elegans*, consistent with their closest relationship observed for birth and death of gene families. We observed seven syntenic blocks between the largest scaffold (5.5 Mb) of *A. avenae* and chromosome III of *C. elegans*, including two inverted syntenic blocks. The longest syntenic block with nineteen genetic linkages spanned 1.76 Mb in *A. avenae* and 1.06 Mb in *C. elegans*. Synteny between *A. avenae* and *C. elegans* suggested an intrachromosomal rearrangement pattern, which is the common theme across the phylum[23].

*A. avenae* gene number is nearly twice that in other sequenced nematodes genomes, likely due to a high level of gene duplication. Compared to gene duplications in one-third of the *C. elegans* genome[24,25], the extent of gene duplications in *A. avenae* increased to 48%. We identified 30 collinearity blocks containing 420 genes (1%) (Supplementary Fig. 4, Supplementary Data 3), as well as 1216 (2.8%), 1146 (2.7%), and 17,942 (41.5%) genes involved in tandem, proximal, and dispersed duplication events,

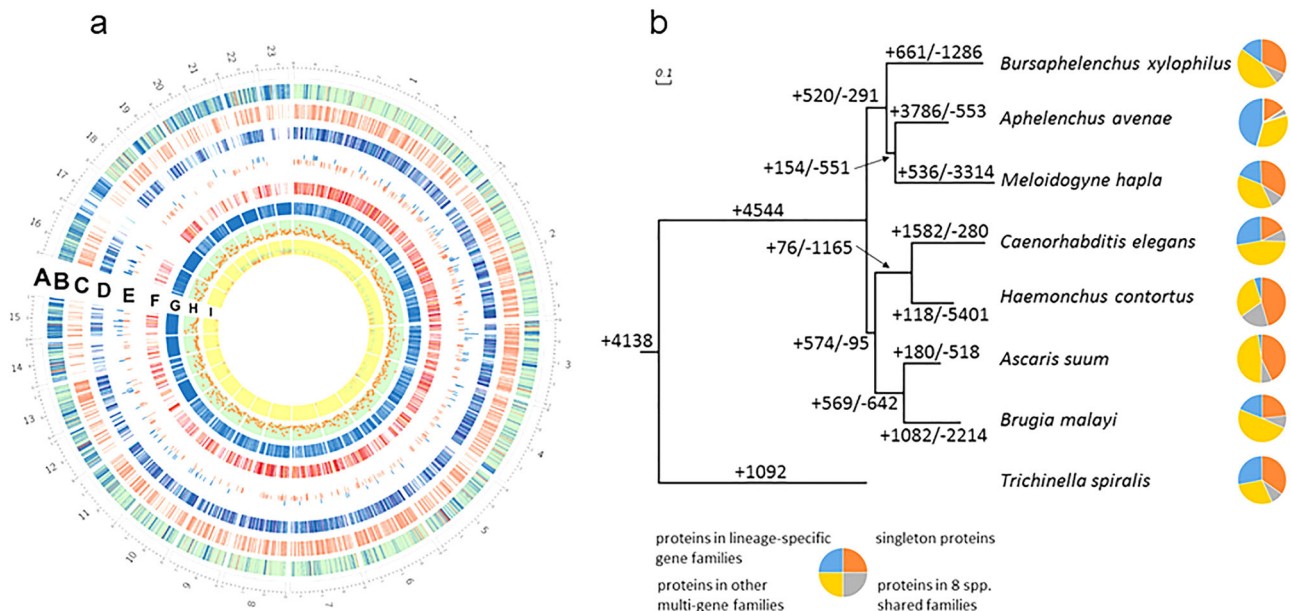

**Fig. 1 Genome features of *A. avenae* genome. a** Plots of the largest 23 scaffolds in *A. avenae* genome. (A) Circular visualization of the largest 23 scaffolds. The axis number stands for the size (Mb) of scaffold span. (B) Gene models on the largest 23 scaffolds in *A. avenae* genome. Red: conserved genes in *A. avenae*, *C. elegans*, *B. malayi*, *M. hapla*, *A. suum*, *B. xylophilus*, *H. contortus*, and *T. spiralis*. Green: additional conserved genes in *A. avenae* and *C. elegans*. Blue: unique genes in *A. avenae*. (C) Predicted operons in *A. avenae* genome. (D) Heatmap of upregulated genes during desiccation stress in *A. avenae*. The color represents the FPKM (fragments per kilobase of transcript length per million mapped reads) values. (E) Histogram of above three-fold change upregulated genes (blue) and downregulated (light red) during desiccation in *A. avenae*. (F) Heatmap of downregulated genes during desiccation in *A. avenae*. The color represents the FPKM values. (G) Repeats in *A. avenae* genome (variations per kb). (H) SNP distribution in *A. avenae* genome. (I) Genome sequencing read coverage (average per 100 bases). Yellow: >100x. Red: <40×. Blue: >40× but <100×. **b** Gene gains and losses across eight nematode species. The numbers of gene gains (+) and losses (−) are listed on branches. Singletons or shared genes across the eight nematode species were shown in pie charts.

respectively. Based on Gene Ontology (GO) analysis, the top three functions of the duplicated genes were assigned to GO terms with hydrolase activity, protein binding, and nucleic acid binding.

**Pathways and genes involved in *A. avenae* anhydrobiosis.** To examine the global gene expressions in response to extreme desiccation, we sequenced the transcriptomes of *A. avenae* prepared in five conditions. The selected conditions include 100% relative humidity (rh, fresh control), 97% rh (35% water content), 85% rh (8% water content), 40% rh (7% water content), and 0% rh (1% water content)[19]. Illumina Hiseq 2000/2500 generated 492 million paired-end reads and at least 43 X coverage per sample. Both principle components analysis and multi-dimensional scaling analysis confirmed that desiccation caused global pattern of differential gene expressions (DGE) (Supplementary Figs. 5 and 6). Comparing DGE in 97% rh condition to fresh-condition, we observed 9211 upregulated and 5260 downregulated genes (q < 0.05) (Fig. 2a, Supplementary Fig. 7). Furthermore, *A. avenae* differentially expressed hundreds of genes in 85, 40, and 0% rh compared to 97% rh. These data suggest that global responses occurred at the early stage of gradual water loss in *A. avenae*, supporting the hypothesis that slow drying may allow commencement of protective mechanisms that limit cell damage[26]. 58 genes, such as those encode small heat shock protein 12.6 and kinases, were differentially expressed in all the conditions (Fig. 2b). Although they are distinct at the pre-condition stage by up and downregulation, all of them were upregulated at 85 and 40% rh and downregulated at 0% rh.

Next, geneset analysis identified 37 upregulated pathways (34%) and 6 downregulated pathways (6%) based on KEGG databases (adjusted *p* ≤ 0.05) (Table 1)[27]. Additionally, pathway

studio-based manual analysis confirmed similar regulation patterns (Fig. 3a, Supplementary Figs. 8–22, Supplementary Note). We observed that *A. avenae* enhanced gene expressions in purine and pyrimidine metabolism, and DNA repair and replication pathways, indicating that *A. avenae* has successfully survived by exploiting chromosome protection mechanisms to withstand water loss. Ubiquitin mediated proteolysis pathway was significantly upregulated to breakdown proteins, which may provide amino acids for protective molecule synthesis. In addition, TCA cycle, oxidative phosphorylation, and electron transfer chain pathways were upregulated for generating more energy currency ATP that may be involved in repairing and recycling of the macromolecules to enter ametabolic state. Majority of genes in autophagy pathway were upregulated, suggesting that it could be an effective way to remove damaged cells and aggregated proteins during water loss. Consistently, upregulated autophagy pathway has been observed in insect, yeast, and resurrection plants[28–30]. In contrast, tardigrade *Ramazzottius varieornatus* genome suppresses autophagy induction[2]. These data indicate species have evolved their unique protective mechanisms against extreme desiccation. Intriguingly, we identified that Notch signaling and endocytosis pathways were upregulated during desiccation. Notch signaling pathway mediates diverse cellular processes during development including cell fate decisions in *C. elegans*[31,32]. Consequently, Notch signal activation requires endocytosis mediated vesicle trafficking for regulating downstream gene expressions. Although the specific functions of Notch signaling and endocytosis pathways involved in regulation of anhydrobiosis remained unknown, our data suggest that Notch signaling may determine cell specialization adapted to desiccation. Taken together, these findings indicate that *A. avenae* undergo dramatic programmed remodeling for entering ametabolic state.

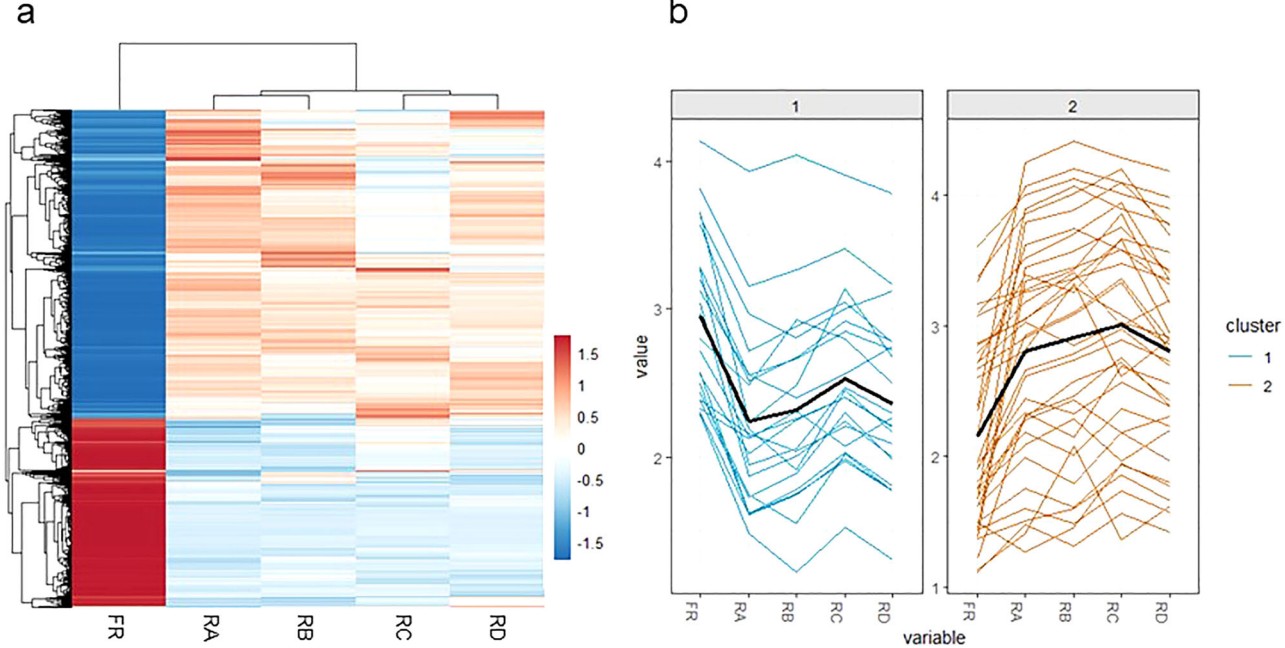

**Fig. 2 Differential gene expression patterns in *A. avenae* during water loss. a** Heatmap of global differential gene expressions in *A. avenae* during water loss. The color represents the z-scores of the expression values of the genes, which were calculated as the mean-centered FPKM values. **b** Expression of 58 DGE across five conditions. FR: 100% rh, RA: 97% rh, RB: 85% rh, RC: 40% rh, RD: 0% rh.

To reveal step-wise transformations of *A. avenae* for entering the anhydrobiotic state, we next examined its DGE patterns between different relative humidity conditions. The expressions of 531 (85% rh vs 97% rh), 1261 (40% rh vs 85% rh), and 665 (0% rh vs 40% rh) genes were significantly upregulated, while those of 359 (85% rh vs 97% rh), 1223 (40% rh vs 85% rh), and 976 (0% rh vs 40% rh) genes were significantly downregulated. Venn diagram plots showed that the up or downregulated genes were rarely overlapped when compared between the relative humidity conditions ranged from 97 to 0% rh (Supplementary Figs. 23 and 24). Moreover, GO analyses indicated that those upregulated genes were involved in hydrolase activity, transferase activity, and molecular binding activity, whereas those downregulated genes played roles in nitrogen compound metabolism, catabolism, and localization (Supplementary Figs. 25 and 26). Thus, as the progression of body water loss occurred, majority of the genes, whose expressions were upregulated at 97% rh, continuously expressed at similar levels, and the expressions of genes that were further upregulated in response to extreme desiccation played roles in macromolecule, phosphorus, lipid, and carbohydrate metabolisms (Supplementary Fig. 25). Consistent with the above conclusions, these data suggest that the massive remodeling processes occur in response to water loss at the early stage of anhydrobiosis (97% rh) and continuously function with additional upregulated metabolism genes for adapting to extreme desiccation.

By manually annotating the top 30 significantly upregulated proteins in desiccation, we found that 21 (70%) proteins were novel putative proteins with unknown functions. Based on disorder protein analysis[33], ten of the 21 novel proteins are either fully disordered or with only a few ordered amino acids. These intrinsically disordered proteins (IDP) are predicted to be able to form alpha-helical structures, thus may have a chaperone like function as LEA and anhydrin during desiccation stress. We further identified 137 proteins are novel IDPs that may function as molecular shields (Supplementary Data 4). Compared to the other seven nematode species, 74 IDPs are species specific in *A. avenae* (Supplementary Data 5).

Like IDPs, LEA proteins may form gel phase to protect organisms from water loss[18,20]. Here we identified fifteen group 3 LEA proteins in *A. avenae* genome (Fig. 3b, Supplementary Note, Supplementary Fig. 27). All of them were significantly upregulated during desiccation, and two of them were ranked in the top-10 significantly upregulated proteins. Neighbor-joining phylogenetic tree revealed that the fifteen *A. avenae* LEAs showed unique protein sequences and were closely clustered (Fig. 3b). Closer phylogenetic relationship of LEAs was inferred between *A. avenae* and plants than *A. avenae* and *C. elegans* (Fig. 3b). These data suggest that dispersed duplications occurred in LEA family of *A. avenae*, facilitating surviving complete desiccation. In addition, horizontal gene transfer of ancient LEA might occur between *A. avenae* and plants.

HSP70 family is another family showed striking gene family expansion in *A. avenae* genome, comparing to six copies in fruit fly, 22–32 in tapeworms, eighteen in *A. thaliana* and two in human[34]. Fifty HSP70 proteins were identified by sequence homolog in *A. avenae* and forty of them had expression values in our transcriptome sets. Six of *A. avenae* HSP70 genes (12%) were duplicated by WGD. Maximum-likelihood-based phylogenetic tree of HSP70 from *A. avenae* and other nematodes suggested that the majority of *A. avenae* HSP70 genes were species specific (Supplementary Fig. 28). Thirteen HSP70 genes were significantly upregulated and three were significantly downregulated during desiccation stress (Supplementary Fig. 28). These differentially expressed HSP70 genes were not clustered together in the phylogenetic tree, indicating sequence diversity of desiccation-resistant HSP70 proteins.

Genome analysis identified a remarkable count of 767 conventional protein kinases (EPKs) and additional nineteen atypical protein kinases (APKs) in *A. avenae* (Supplementary Table 10), representing the second largest reported metazoan kinome next to *Haemonchus contortus*[35]. The kinome expansions occurred in AGC group (cAMP-dependent, cGMP-dependent and protein kinase C) and two APK groups: phosphatidylinositol 3-kinase-related kinases (PIKK) and "right open reading frame" (RIO) (Supplementary Note). Transcriptome analysis revealed 230 upregulated and 126 downregulated kinase genes at 97% rh

**Table 1 KEGG pathway enrichment of differentially regulated genes during anhydrobiosis.**

| (a) Upregulated KEGG pathway enrichment pathway | Upregulated gene number | | | | Total gene number |
|---|---|---|---|---|---|
| | 97% vs 100% | 85% vs 100% | 40% vs 100% | 0% vs 100% | |
| Protein processing in endoplasmic reticulum | 133 | 131 | 124 | 128 | 149 |
| Spliceosome | 116 | 119 | 113 | 117 | 127 |
| Ribosome | 119 | 118 | 114 | 116 | 121 |
| RNA transport | 103 | 106 | 98 | 103 | 109 |
| Purine metabolism | 83 | 81 | 80 | 82 | 102 |
| Ubiquitin mediated proteolysis | 88 | 88 | 83 | 85 | 96 |
| Oxidative phosphorylation | 94 | 94 | 89 | 92 | 95 |
| Endocytosis | 70 | 69 | 66 | 66 | 88 |
| Ribosome biogenesis in eukaryotes | 66 | 63 | 65 | 65 | 78 |
| Phagosome | 66 | 68 | 66 | 64 | 74 |
| Pyrimidine metabolism | 64 | 61 | 60 | 62 | 70 |
| mRNA surveillance pathway | 60 | 61 | 59 | 57 | 62 |
| Peroxisome | 45 | 43 | 42 | 41 | 53 |
| Proteasome | 50 | 51 | 50 | 51 | 50 |
| Amino sugar and nucleotide sugar metabolism | 41 | 35 | 46 | 42 | 49 |
| DNA replication | 47 | 42 | 44 | 48 | 47 |
| Aminoacyl-tRNA biosynthesis | 42 | 40 | 41 | 41 | 43 |
| RNA degradation | 39 | 41 | 39 | 41 | 42 |
| Citrate cycle (TCA cycle) | 32 | 32 | 33 | 32 | 41 |
| Nucleotide excision repair | 36 | 34 | 34 | 35 | 37 |
| N-Glycan biosynthesis | 28 | 26 | 27 | 22 | 32 |
| Basal transcription factors | 31 | 33 | 31 | 32 | 32 |
| Fanconi anemia pathway | 29 | 27 | 27 | 27 | 29 |
| Base excision repair | 27 | 26 | 26 | 27 | 27 |
| Homologous recombination | 22 | 22 | 22 | 21 | 22 |
| RNA polymerase | 21 | 19 | 17 | 18 | 22 |
| Mismatch repair | 20 | 19 | 18 | 20 | 20 |
| Pentose phosphate pathway | 18 | 19 | 18 | 18 | 20 |
| Protein export | 17 | 16 | 16 | 16 | 19 |
| Galactose metabolism | 15 | 15 | 15 | 15 | 16 |
| Terpenoid backbone biosynthesis | 14 | 14 | 13 | 14 | 15 |
| Notch signaling pathway | 13 | 12 | 12 | 11 | 14 |
| 2-Oxocarboxylic acid metabolism | | | 10 | | 11 |
| Non-homologous end-joining | 8 | 7 | 7 | 8 | 8 |
| Glycosylphosphatidylinositol(GPI)-anchor biosynthesis | 8 | | | | 8 |
| Ubiquinone and other terpenoid-quinone biosynthesis | | | 8 | | 8 |
| Nicotinate and nicotinamide metabolism | 7 | | | | 7 |
| Synthesis and degradation of ketone bodies | 6 | 6 | | 6 | 6 |

| (b) Downregulated KEGG pathway enrichment pathway | Downregulated gene number | | | | Total gene number |
|---|---|---|---|---|---|
| | 97% vs 100% | 85% vs 100% | 40% vs 100% | 0% vs 100% | |
| Drug metabolism—cytochrome P450 | 55 | 55 | 58 | 59 | 76 |
| Metabolism of xenobiotics by cytochrome P450 | 38 | 40 | 41 | 41 | 52 |
| Calcium signaling pathway | 38 | 37 | 38 | 43 | 50 |
| Cysteine and methionine metabolism | 23 | 28 | 25 | 25 | 36 |
| Retinol metabolism | | 19 | 17 | 18 | 26 |
| Neuroactive ligand-receptor interaction | 18 | 19 | 17 | 18 | 19 |
| Histidine metabolism | | | 10 | 10 | 13 |
| Taurine and hypotaurine metabolism | 4 | | 4 | | 4 |

compared to 100% rh condition (Supplementary Data 7), implying that phosphorylation network senses water loss and transfers signals to regulate functional gene expressions. It was reported that overexpression of a plant homologue of the yeast AGC kinase, Dbf2, enhances drought tolerance in yeast as well as in transgenic plant cells[36]. Thus, the *A. avenae* kinome expansion may result from evolutionary adaptation to repetitive water loss.

## Conclusion
In this study, the genome and transcriptome of *A. avenae* provide molecular insights into metazoan survival of extreme desiccation.

Recent genome-wide duplication may account to the unexpected large genome size of *A. avenae*, which benefits *A. avenae* adaption to repeated dehydration and rehydration cycles for millions of years. The expanded and diversified kinome represents the second largest metazoan kinome, which may play roles in explicitly transducing signals in response to extreme desiccation stress. We showed that *A. avenae* reprogrammed its global metabolisms for entering anhydrobiosis. Moreover, 74 species-specific IDPs which may serve as potential molecular shields for extreme desiccation resistance were identified. Further functional characterization of these protective molecules will provide insights to desiccation survival.

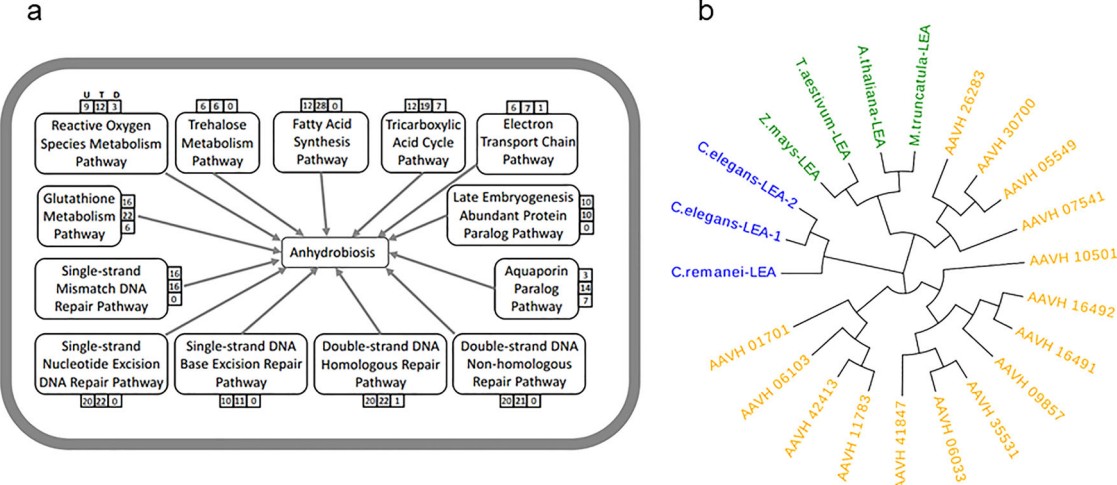

**Fig. 3 Essential pathways and duplication of LEA proteins involved in *A. avenae* anhydrobiosis. a** Schematic of pathways involved in *A. avenae* anhydrobiosis. U: upregulated gene number, D: downregulated gene number, T: total number of DGE genes. **b** Phylogenetic tree of group 3 LEA proteins in *A. avenae*, *C. elegans*, and *A. thaliana*.

## Methods

**Genomic DNA isolation and raw data generation.** *Aphelenchus avenae* was reared on plant pathogenic fungus *Rhizoctonia solani* reared on a substrate of autoclaved wheat[19]. The nematodes were washed using distilled water, collected on filter, and then harvested by 30% sucrose float to remove contaminants. Aliquots of nematodes were preserved at −80 °C. To isolate the genomic DNA, the nematodes were lysed by lysis buffer, containing 0.1 M Tris-Cl pH 8.5, 0.1 M NaCl, 50 mM ethylenediaminetetraacetic acid (EDTA) pH 8.0, 1% sodium dodecyl sulfate (SDS), treated with Protease K. Then the phenol/chloroform extraction method was carried out. The 18 S ribosomal RNA gene was amplified using polymerase chain reaction (PCR) and identified by Sanger sequencing on an Applied Biosystems (ABI) 3730 XL DNA Analyzer. No contamination was found in the sequencing results of the 18 S ribosomal RNA gene.

Roche 454 shotgun libraries and 8-kb and 20-kb span paired-end libraries were prepared based on Roche 454 manual and sequenced on Roche GS FLX titanium platform at University of Hawaii. Illumina 500 bp paired-end library was prepared based on Illumina manual and sequenced on Illumina Hiseq 2000 platform at Yale Center for Genome Analysis (YCGA).

**Genome assemblies.** The Illumina reads were scanned and trimmed to remove poor quality bases using Trimmomatic v0.36[37]. Since the sequencing errors generated by Illumina platform is 0.5–2.5%[38], the Illumina reads were then corrected by Quake v0.2 which uses a maximum-likelihood approach and parallel K-mer counting method[39]. Both flow cytometry method and K-mer frequency distribution method were applied to estimate the genome size (Supplementary Note). Genome assembly reported here was performed using SOAPdenovo v2 and Newbler v2.8, and then Illumina paired-end reads were mapped to close gaps (Supplementary Note). The accuracy of the assembly was evaluated by SOAPaligner v1 package. The completeness of the assembly was evaluated by a core eukaryotic genes mapping approach (CEGMA v2.5) and BUSCO v5.0.0[21] (Supplementary Note).

**RNA isolation and RNA-seq.** *A. avenae* samples, containing three biological replicates for each condition, were prepared at 100, 97, 85, 40, and 0% relative humidity (R.H.). All the samples prepared for desiccation-stress conditions were preconditioned at 97% relative humidity (rh) for 72 h. For each condition, more than 1 mg of mixed-stage nematodes were collected. Ten volumes of Trizol per sample was added. Samples were vortexed briefly, and frozen in dry ice-ethanol bath, and preserved at −80 °C until isolation. For total RNA isolation, nematode samples were thawed at 37 °C, and then placed in dry ice-ethanol bath, which was repeated for six times. Then the nematode samples were placed on ice for 30 s and vortexed for 30 s, which were repeated for six times. After the nematode samples were kept at room temperature for 5 min, 2 ml of chloroform per ml was added. The samples were shaken for 15 s, and then sit at room temperature for 2–3 min. The samples were centrifuged at 4 °C at $12,000 \times g$ for 15 min. Aqueous phase was transferred to a new tube, and 1 volume of isopropanol was added, and incubated at room temperature for 10 min. RNA was pelleted at $12,000 \times g$ for 10 min. RNA pellets were rinsed with 75% ethanol, centrifuged at $7500 \times g$ for 5 min. Ethanol was removed, and RNA pellets were dissolved with diethylpyrocarbonate (DEPC) treated water. RNA quality was evaluated by nanodrop and Agilent 2100 bioanalyzer with RNA Pico chips.

Messenger RNA (mRNA) was isolated and cDNA libraries were prepared according to Illumina mRNA sequencing manual. Barcodes containing 6 bp were ligated to each sample, and 15 samples were pooled together for Illumina 75 bp paired-end sequencing on Hiseq 2000 and 2500 platforms. A total of 492 million paired-end reads were generated. Detailed summaries of reads were listed in Supplementary Table 5.

**Repeat, transposon, and noncoding RNA annotation.** To identify de novo repeat families in *A. avenae* genome, we used RepeatModeler v1.03[40] combining de novo repeat finder RECON v1.05[41], de novo repeat finder RepeatScout v1.05[42], tandem repeat finder TRF v4.09[43], and NCBI RMBLAST v2.2.27+. The assembled scaffolds were searched by running transposonPSI v20100822 via PSI-TBLASTN to identify transposon families in *A. avenae* genome. By combining Repbase library and outputs from RepeatModeler and transposonPSI, RepeatMasker v4.06 program[40] was used to identify 16.7% of *A. avenae* genome as repetitive and transposon regions. The details of repetitive families were listed in Supplementary Table 3.

**Protein-coding gene annotation.** Two sets of data were prepared to train AUGUSTUS v2.5.5[44] for gene model prediction in *A. avenae* genome. One set was generated based on PASA v2.0.2 validated transcripts, while the other set was generated from eukaryotic orthologous groups (KOGs) identified in *A. avenae* genome by CEGMA v2.5. The evaluation of accuracy reports showed that AUGUSTUS trained by KOG set generated better accuracy results at nucleotide, exon, and gene levels (Supplementary Table 6). Thus we used KOG set trained parameter to predict gene models by AUGUSTUS. We further used *A. avenae* KOG set to train SNAP v20131129 and GlimmerHMM v3.02[45] for gene model prediction. GeneMark-ES[46] was trained by *A. avenae* genome sequence, since GeneMark-ES v4.38 calculates species-specific parameters for gene predictions from DNA sequence itself by using self-training algorithm, without needing extra species-specific transcript information for training. Then the genesets predicted by four ab initio programs and the high-quality transcripts validated by PASA pipeline were combined for final genesets generation[47]. Operon analysis was carried out using in-house perl script counting genes located on the same strand and had intergenetic regions ranging from 25 to 1000 bases (Supplementary Note).

**Ortholog cluster analysis.** To identify clusters of putative orthologs (COGs), we performed stand-alone OrthoMCL v2.0.9 software[48], which employs a Markov Cluster algorithm and groups proteins based on sequence similarity. OrthoMCL firstly removed low quality sequences. All of 43,192 *A. avenae* proteins passed criteria and were kept for downstream steps. An all-vs-all BLASTP analysis was performed on proteome sets by using NCBI BLASTP with 1e-5 as e value cutoff. Then all-vs-all BLASTP output was loaded to generate a MySQL database. Putative orthologs with best similarity scores were identified between species while paralogs with better similarity scores were identified within species.

Protein sequences from *C. elegans*, *B. malayi*, *M. hapla*, *A. suum*, *B. xylophilus*, *H. contortus* and *T. spiralis* were downloaded from wormbase website (https://wormbase.org/) as WS233 version.

**Phylogenetic tree and gene gain/loss.** To build the phylogenetic tree of eight nematode species, 168 clusters of single-copy orthologs from eight nematode genomes were aligned by MAFFT v7.046b program[49] and then concatenated for

tree estimation by RAxML v 8.1.3 package[50]. *T. spiralis* was set to be the out group. Gamma model of rate heterogeneity and DAYHOFF model for amino acid substitution matrix were applied. Then 1000 rapid bootstrap inferences were executed for large scale maximum-likelihood analyses. Gene gains/losses were analyzed by Dollo and Polymorphism Parsimony program (Dollop) of the PHYLIP package v3.695[51]. Detailed methods for the phylogenetic tress of LEA and HSP70 proteins appear in Supplementary Note.

**Genome duplication, collinearity, and synteny**. To identify paralogs in *A. avenae* genome, we performed all-vs-all BLASTP analysis on 43,192 gene models. To further examine pair-wise collinear relationships and other type gene duplications, we performed MCScanX program[52] to scan NCBI BLASTP output and cluster collinear regions. The collinear relationships in the largest 23 scaffolds were visualized by Circos v0.64[53]. Collinear gene list was shown in Supplementary Data 3. Syntenic relationships were analyzed by SyMAP 4.2[54,55] and then presented by Circos v0.64.

**Transcriptome assembly and differentially expressed genes (DEGs) analysis**. RNA-seq reads were mapped to the assembled scaffolds, and exons and possible splice junctions were identified by TopHat2[56] through bowtie2 v2.1.0[57]. TopHat2 collected initial mapping and potential exons information to create a possible splice junction database. Then all the input reads were splitted into smaller fragments and mapped independently again, to avoid missing exons smaller than 100 bp. TopHat2 pooled smaller segment alignment information together and generated an end-to-end read alignments in the final step. Cufflinks2[58,59] was then used to assemble TopHat2 output to transcripts. The assembled transcripts were realigned to the draft genome assembly by a genomic mapping and alignment program (GMAP) and validated by program to assemble spliced alignments (PASA) pipeline. Comparative transcriptome analysis was done using Cuffdiff. Further statistical analyses were performed by R package CummeRbund v2.33.0[60]. Heatmap graphs of DGEs were generated by Cluster 3.0 and visualized by Java TreeView v1.1.6. We further identified Kyoto encyclopedia of genes and genomes (KEGG) pathways and reactome pathways by running locally installed KOBAS 2.0[61]. For geneset analysis (GSA), we used R package piano v2.6.0[62].

**Statistics and reproducibility**. Statistical analyses were carried out using Cufflinks v2, CummeRbund v2.33.0 and piano v2.6.0. *P*-values <0.05 were considered significant. For multiple hypothesis testing, adjusted *p*-values <0.05 were considered significant. Three biological replicates for each condition were performed.

**Reporting summary**. Further information on research design is available in the Nature Research Reporting Summary linked to this article.

## Data availability

The supplementary data required to generate the images and interpret the results of this study are openly available in Figshare at https://doi.org/10.6084/m9.figshare.16640293[63]. The sequencing data are deposited in the Sequence Read Archive database (PRJNA236621, PRJNA236622). The genome assembly with gene annotations is deposited under PRJNA236621. All data are available in the manuscript or the supplementary materials.

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

## Acknowledgements
This work was supported by Advanced Studies in Genomics, Proteomics and Bioinformatics, University of Hawaii (M.A.). USDA is an equal opportunity employer. Mention of trade names or commercial products in this publication is solely for the purpose of providing specific information and does not imply recommendation or endorsement by the USDA.

## Author contributions
X.W., J.A.S., and M.A. designed the study and wrote the manuscript. X.W., S.H., L.M.H., and C.Z.W. performed the experiments, X.W., J.A.S., S.M.G., and A.Y. analyzed the data. M.A. acquired funding.

## Competing interests
The authors declare no competing interests.
