## [Transparent Peer Review File · Communications Biology]

Reviewers' Comments:

Reviewer #1:

Remarks to the Author:

The manuscript, "Life Without Water: the *Aphelenchus avenae* Genome Highlights Evolutionary Adaptation to Desiccation" presents details of work sequencing and analyzing the genome and comparative transcriptomes from a desiccation tolerant nematode.

A basic overview of the genome is performed, highlighting the fact that many unique genomes are found within the genome of this particularly hardy nematode.

For me the most interesting analysis comes from the transcriptome sequencing and comparisons at different relative humidity. One interesting finding is that initial drying at 98% humidity results the major transcriptomic changes. However, even if this is the case it would have been nice to see more comparisons between the different drying regimes. The major pathways found to be upregulated are all the usual things one would expect to see. It would have been nice to see the authors dig into some of the unknown genes a bit more.

Overall the paper is well written, but an expanded analysis of all transcriptomic conditions to one another should be performed. GO enrichment or a similar approach to directly compare *A. avenae* to *C. elegans* would also be good, since much is known about how *C. elegans* senses and responds to drying stress.

Reviewer #2:

Remarks to the Author:

The manuscript "Life Without Water: the *Aphelenchus avenae* Genome Highlights Evolutionary Adaptation to Desiccation" by Wan et al. presents the genome of a cryptobiotic and parthenogenetic nematode. The authors analyse the gene composition of the genome and conduct a RNA-Seq analysis of roundworms during the desiccation process.

The manuscript is well written and I only see one major issue. This pertains the very high number of genes. The number of 26,537 proteins unique to *A. avenae* is larger than the gene number of e.g. *C. elegans* or *Panagrolaimus* species. This seems unlikely and it is well possible that there is a considerable over-prediction of genes. One possible reason for this could be polyploidy, which is found frequently in parthenogenetic nematodes. The authors should check their data in this regard.

In particular, a over prediction of genes will have influenced downstream analysis, like the one on gene duplications and the number of kinases or Leas found. This needs to be checked accordingly.

Minor comments:

- The genome completeness should be reported with BUSCO values, not the outdated CEGMA metric.
- Was the genome data screened against *Rhizoctonia solani* reads/contigs? This is crucial to know and needs to be done.

I hope this review will help the authors to further improve their manuscript.

Best regards

Philipp Schiffer

Reviewer #3:

Remarks to the Author:

This work generates an assembled genome of *Aphelenchus avenae* with protein-coding gene annotations and finds out genes/pathways up or down-regulated under different desiccated conditions. The results in this work provide a genomic/transcriptomic foundation to explain the mechanisms of anhydrobiosis and reveal the desiccation tolerance of metazoa based on this mycophagous nematode.

I am excited to see this paper as a nematologist working on adaptations of nematodes under extreme environmental stressors. Generally, this is a nice work with informative content, the results of genome and transcriptome are acceptable to me. However, I do have a few concerns that I listed below with my suggestions:

Page 2 line 20: The author stated "complete body water" which I am not sure about this, I know the most tolerance of metazoa is around losing 98% of the body water, so make sure you have solid evidence here.

Page 2 line 21: Not few eukaryotes that do anhydrobiosis I believe. Numbers of nematodes can do this such as *Plectus*, *Panagrolaimus*, *Scottinema*.....

Page 3 line 44: I would love to know what are those three kingdoms of life

Page 3 line 46: you could be more specific when you mention "certain stage of life cycle", such as young adult without reproduction? Also, you need citations here!

Page 3 line 46: "few" or "a few"? Or a limited number of ?? Anyway, I think "few" is not appropriate.

Page 3 line 47: Not only the adult stage, I think Antarctic nematodes can tolerate extreme desiccation (or employ anhydrobiosis) in both larva and adult stages.

Page 3 line 53: You should have citations about Antarctic nematodes that can tolerance desiccation and freezing in this paragraph since you are talking about life at the extreme. Check papers from Adhikari (2009, 2010), Wharton (2002, 2005, 2015) Xue (2021), which may be helpful for your instruction.

Page 3 line 57-58: How about *Plectus murrayi* that is *Spirulina*? Also, you do need citations here!

Page 3 line 59: Citations needed.

Page 3 line 61: I won't use "evolved" here, maybe developed.

Page 3 line 62: Citations needed.

Page 4 line 66: Again, I suggest the authors do more literature search for the introduction, I do believe Adhikari provides some cool stuff from his 2009 and 2010 work.

Page 4 line 75: It will be great if you could add the quality control information in your result section.

Page 4 line 77: Mb refers to byte or base?

Page 4 line 81: The number of scaffolds is large with the N50 is fine, have you ever try anything to get a better assembly?

Page 4 line 84: The Chinese title in supplementary figure 2 should be changed to English for more readers.

Page 5 line 92: You can evaluate the completeness of your assembled genome by using BUSCO (Nematoda or metazoa as reference)

Page 6 line 113: Why you used the largest 23 scaffolds in the genome? Because they are more informative or other reasons? I did not see any discussion about this in your discussion.

Page 6 line 123: What is the lowest coverage? 10X?

Page 8 line 179: keep nematodes in different water content for how long?

Page 10 figure 2: It is impossible to see the genes, so I cannot get any information from this figure, do you mean only expression pattern?

Page 10 line 213-215: I will be very careful to draw this conclusion. It is strange to say "successfully evolved" by only detecting the expression level of some certain genes that associated with certain functions. Besides, "successfully evolved" itself is not really appropriate, we would use "successfully survived".

Page 12 figure 3: Interesting results, maybe make the node labels bigger, it is hard to read.

Page 13 line 284: Citations needed!

Page 13 line 290: If you try to explain horizontal gene transfer, maybe not only use neighbor-joining for tree construction, and why you change to ML in HSP70?

Page 14 line 319: This section looks definitely not a discussion, it is more like a conclusion. I found a lot of discussion in your result section actually. I suggest you reorganize your result and discussion. Of course, with necessary citations in the text.

Page 15 line 334: Are the nematodes cultured on the medium or other materials?

Page 15 line 336: So you mean the sucrose float is for all contaminants removal? Including fungus, right?

Page 16 line 348: Trimmed with what program? Trimmomatic?

Page 16 line 353: Citations for soap and newbler.

Page 16 line 356: I have not find the results from CEGMA.

Page 16 line 359: each RH for how long?

Page 19 line 412-413: It will be better to add the website or FTP link here.

Page 19 line 414: Need more information for phylogeny construction.

Page 20 line 426: So largest 16 scaffolds instead of 23, why?

Page 21 line 425: NCBI, right?

Page 21 Reference: Some of them need formatted

Reviewers' comments:

Reviewer #1 (Remarks to the Author):

The manuscript, "Life Without Water: the *Aphelenchus avenae* Genome Highlights Evolutionary Adaptation to Desiccation" presents details of work sequencing and analyzing the genome and comparative transcriptomes from a desiccation tolerant nematode.

A basic overview of the genome is performed, highlighting the fact that many unique genomes are found within the genome of this particularly hardy nematode.

For me the most interesting analysis comes from the transcriptome sequencing and comparisons at different relative humidity. One interesting finding is that initial drying at 98% humidity results the major transcriptomic changes. However, even if this is the case it would have been nice to see more comparisons between the different drying regimes. The major pathways found to be upregulated are all the usual things one would expect to see. It would have been nice to see the authors dig into some of the unknown genes a bit more.

Overall the paper is well written, but an expanded analysis of all transcriptomic conditions to one another should be performed. GO enrichment or a similar approach to directly compare *A. avenae* to *C. elegans* would also be good, since much is known about how *C. elegans* senses and responds to drying stress.

Response: Thank you for your suggestions. We have added more information regarding the comparisons between the different drying regimes in the main text. Please see the text below (lines 268-285 in the revised manuscript and Supplementary Figures 23-26).

“To reveal step-wise transformations of *A. avenae* for entering the anhydrobiotic state, we next examined its DGE patterns between different relative humidity conditions. The expressions of 531 (85% rh vs 97% rh), 1,261 (40% rh vs 85% rh), and 665 (0% rh vs 40% rh) genes were significantly up-regulated, while those of 359 (85% rh vs 97% rh), 1,223 (40% rh vs 85% rh), and 976 (0% rh vs 40% rh) genes were significantly down-regulated. Venn diagram plots showed that the up- or down-regulated genes were rarely overlapped when compared between the relative humidity conditions ranged from 97% rh to 0% rh (**Supplementary Figures 23 and 24**). Moreover, GO analyses indicated that those up-regulated genes were involved in hydrolase activity, transferase activity, and molecular binding activity, whereas those down-regulated genes played roles in nitrogen compound metabolism, catabolism, and localization (**Supplementary Figures 25 and 26**). Thus, as the progression of body water loss occurred, majority of the genes, whose expressions were upregulated at 97% rh, continuously expressed at similar levels, and the expressions of genes that were

further upregulated in response to extreme desiccation played roles in macromolecule, phosphorus, lipid, and carbohydrate metabolisms (**Supplementary Figure 25**). Consistent with the above conclusions, these data suggest that the massive remodeling processes occur in response to water loss at the early stage of anhydrobiosis (97% rh) and continuously function with additional up-regulated metabolism genes for adapting to extreme desiccation.”

We agree that a direct comparison between *A. avenae* and *C. elegans* will be informative. If we consider the comparisons at transcriptomal levels, specific differences between experimental conditions need to be considered. The detailed comparisons can be done in future works and prepared for a new manuscript.

Reviewer #2 (Remarks to the Author):

The manuscript "Life Without Water: the *Aphelenchus avenae* Genome Highlights Evolutionary Adaptation to Desiccation" by Wan et al. presents the genome of a cryptobiotic and parthenogenetic nematode. The authors analyse the gene composition of the genome and conduct a RNA-Seq analysis of roundworms during the desiccation process.

The manuscript is well written and I only see one major issue. This pertains the very high number of genes. The number of 26,537 proteins unique to *A. avenae* is larger than the gene number of e.g. *C. elegans* or *Panagrolaimus* species. This seems unlikely and it is well possible that there is a considerable over-prediction of genes. One possible reason for this could be polyploidy, which is found frequently in parthenogenetic nematodes. The authors should check their data in this regard.

In particular, an over prediction of genes will have influenced downstream analysis, like the one on gene duplications and the number of kinases or Leas found. This needs to be checked accordingly.

Response: Thank you for your suggestions. Estimating the genome size is important to determining the quality of the assembly at the beginning stage of the project. Thus, we applied two independent methods, flow cytometry experiment and k-mer analysis, to estimate the genome size of *A. avenae*. Please see the method details described in the supplementary note. Both methods suggest that the genome size of *A. avenae* is ~255 Mb, which is consistent with the size of the assembled genome. In addition, our flow cytometry analysis suggested that *A. avenae* has a diploid genome. We next analyzed the genome-wide SNP in *A. avenae* and found that the mean SNP density was about 3.13 variants per kb (0.3%). The genome-wide SNP is estimated to range from ~3.3 to 10 variants per kb in any genome (DOI 10.1186/s13059-015-0678-1). Thus, the genome-wide SNP rate in *A. avenae* falls to this standard range towards the low end.

Because SNP rates within polyploid genomes will show polymorphism between homologous sequences, the values will be expected to be much higher. Taken together, these data suggest that the sequenced *A. avenae* genome is not a polyploidy genome.

In addition, the protein-coding sequence percentage (% of genome) of *A. avenae* is 15%, which is similar to that of *B. malayi* (17.8%) and lower than that of *M. incognita* (25.3%). Moreover, the gene density (genes per Mb) of *A. avenae* (169) is similar to that of *B. malayi* (162) and lower than those of *M. incognita* (223) and *C. elegans* (260). Thus, we consider the predicted gene numbers are moderate, because the genome size of *A. avenae* is at the high end of the nematode genome size spectrum. When we further consider that the genome size of *A. avenae* is 2.5 times of the size of *C. elegans*, and there are gene gain-and-loss events during genome evolution, the high unique gene number may be surprising but not unacceptable.

Minor comments:

- The genome completeness should be reported with BUSCO values, not the outdated CEGMA metric.

Response: According to your suggestions, we used BUSCO v5.0.0 to assess the genome assembly. Using eukaryote_odb10 dataset (255 genes, 70 species), we identified 92.6% of the BUSCO orthologs in the assembly. Using metazoan_odb10 (954 genes, 65 species), we identified 77% of the BUSCO orthologs in the assembly. We also used nematode_odb10 dataset (3,131 genes, 7 species), we identified 73.6% of the BUSCO orthologs in the assembly. In addition, BUSCO does not perform uniformly across all eukaryotic clades (<https://doi.org/10.1186/s13059-020-02155-4>) and the outputs may be affected by some factors. We present the best result from eukaryote dataset in the main text. Please see lines 93-95 in the main text. We also added the information for all the datasets to the supplementary Note (lines 96-99).

- Was the genome data screened against *Rhizoctonia solani* reads/contigs? This is crucial to know and needs to be done.

Response: According to your suggestions, we downloaded the sixteen current available genomes of *Rhizoctonia solani* from the NCBI database and screened them against the assembled *A. avenae* genome using BLASTN (cutoff E_value: 1e-5). Only ~180 average hits per *R. solani* genome (ranged from 23 hits to 264 hits) were found. The hits with aligned lengths above 1 kb were identified as 18S ribosomal RNA homologs (percentages of identical matches: 75.6% ~ 77.8%). The hits with aligned lengths ranged from 100 bp to 600 bp were identified as parts of 18S, 28S ribosomal RNA homologs or highly conserved mRNA sequences that were also found in other species by manual BLASTN searches at the NCBI website. The majority of the hits showed that the aligned lengths were less than 90 bp (percentages of identical matches: < 100%). These data confirm that there are no contaminations from *R. solani* DNA sequences in the assembled *A. avenae* genome.

The GenBank accession numbers of the sixteen *R. solani* genomes were listed as the following: GCA_016906535.1, GCA_017311305.1, GCA_015342405.1, GCA_015342435.1, GCA_015341985.1, GCA_015342415.1, GCA_000715385.1, GCA_000832345.2, GCA_000524645.1, GCA_000334115.1, GCA_001286725.1, GCA_900185085.1, GCA_000695385.1, GCA_001899475.2, GCA_000350255.1, GCA_003268435.1. We have revised the supplementary information accordingly. Please see lines 71-87 in the Supplementary Note.

Reviewer #3 (Remarks to the Author):

This work generates an assembled genome of *Aphelenchus avenae* with protein-coding gene annotations and finds out genes/pathways up or down-regulated under different desiccated conditions. The results in this work provide a genomic/transcriptomic foundation to explain the mechanisms of anhydrobiosis and reveal the desiccation tolerance of metazoa based on this mycophagous nematode.

I am excited to see this paper as a nematologist working on adaptations of nematodes under extreme environmental stressors. Generally, this is a nice work with informative content, the results of genome and transcriptome are acceptable to me. However, I do have a few concerns that I listed below with my suggestions:

Page 2 line 20: The author stated “complete body water” which I am not sure about this, I know the most tolerance of metazoa is around losing 98% of the body water, so make sure you have solid evidence here.

Response: Thank you for your suggestions. We added “losing up to 99% of body water” (Crowe JH, Madin KAC: Anhydrobiosis in nematodes: Evaporative water loss and survival. *J Exp Zool.* 1975, 193: 323-34. 10.1002/jez.1401930308.) to explain “complete body water”. Please see line 20.

Page 2 line 21: Not few eukaryotes that do anhydrobiosis I believe. Numbers of nematodes can do this such as *Plectus*, *Panagrolaimus*, *Scottinema*.....

Response: Both “few” and “a few” mean a small number. We wish to use “few” to emphasize the negative side of the meaning (not many).

Page 3 line 44: I would love to know what are those three kingdoms of life

Response: The three kingdoms of life are classified to Archaea, Bacteria, and Eukarya. Most bacteria and archaea (if not all) can be freeze dried in powder form for storage and distribution. They can be looked up in the ATCC website.

Page 3 line 46: you could be more specific when you mention “certain stage of life cycle”, such as young adult without reproduction? Also, you need citations here!

Response: We have added “(e.g. metazoan larvae or/and adult stage)” to line 46 and the references.

Page 3 line 46: “few” or “a few”? Or a limited number of ?? Anyway, I think “few” is not appropriate.

Response: Because less species can tolerate extreme desiccation in adult stages, we wish to use “few” to emphasize the negative side of the meaning (not many).

Page 3 line 47: Not only the adult stage, I think Antarctic nematodes can tolerate extreme desiccation (or employ anhydrobiosis) in both larva and adult stages.

Response: Some species can only tolerate extreme desiccation in larva stage but not adult stage. We wish to say a limited number of species can tolerate extreme desiccation in adult stages.

Page 3 line 53: You should have citations about Antarctic nematodes that can tolerance desiccation and freezing in this paragraph since you are talking about life at the extreme. Check papers from Adhikari (2009, 2010), Wharton (2002, 2005, 2015) Xue (2021), which may be helpful for your instruction.

Response: Because the first paragraph aims to give a general introduction to metazoan anhydrobiosis and the second paragraph aims to introduce anhydrobiosis in Nematoda, we have added the recommended references to lines 58-61 in the second paragraph.

Page 3 line 57-58: How about *Plectus murrayi* that is *Spirulina*? Also, you do need citations here!

Response: We have added the references. Please see lines 58-61.

Page 3 line 59: Citations needed.

Response: We have added the references. Please see lines 62-63 and reference 18.

Page 3 line 61: I won't use “evolved” here, maybe developed.

Response: We have revised this. Please see line 64.

Page 3 line 62: Citations needed.

Response: We have added the references. Please see lines 65-66 and reference 19.

Page 4 line 66: Again, I suggest the authors do more literature search for the introduction, I do believe Adhikari provides some cool stuff from his 2009 and 2010 work.

Response: We have added the recommended references to lines 58-61.

Page 4 line 75: It will be great if you could add the quality control information in your result section.

Response: Thank you for your suggestions. We wish to maintain the main text as concise as possible. Thus, we keep the quality control information in the supplementary note.

Page 4 line 77: Mb refers to byte or base?

Response: Mb refers to base (1,000,000 bp).

Page 4 line 81: The number of scaffolds is large with the N50 is fine, have you ever try anything to get a better assembly?

Response: We tried to use different strategies. This is the best assembly we can get.

Page 4 line 84: The Chinese title in supplementary figure 2 should be changed to English for more readers.

Response: We revised the title in the supplementary figure 2 as the following: “The depth of sequencing coverage across the assembled *A. avenae* genome.”.

Page 5 line 92: You can evaluate the completeness of your assembled genome by using BUSCO (Nematoda or metazoa as reference)

Response: According to your suggestions, we used BUSCO v5.0.0 to assess the genome assembly. Using eukaryote_odb10 dataset (255 genes, 70 species), we identified 92.6% of the BUSCO orthologs in the assembly. Using metazoan_odb10 (954 genes, 65 species), we identified 77% of the BUSCO orthologs in the assembly. We also used nematode_odb10 dataset (3,131 genes, 7 species), we identified 73.6% of the BUSCO orthologs in the assembly. In addition, BUSCO does not perform uniformly across all eukaryotic clades (<https://doi.org/10.1186/s13059-020-02155-4>) and the outputs may be affected by some factors. We present the best result from eukaryote

dataset in the main text. Please see lines 93-95 in the main text. We also added the information for all the datasets in the supplementary Note (lines 96-99).

Page 6 line 113: Why you used the largest 23 scaffolds in the genome? Because they are more informative or other reasons? I did not see any discussion about this in your discussion.

Response: Correct. We used the largest 23 scaffolds to represent the genome information, because they are informative.

Page 6 line 123: What is the lowest coverage? 10X?

Response: The lowest coverage is 17X.

Page 8 line 179: keep nematodes in different water content for how long?

Response: For 24 hrs (Higa and Womersley, 1993).

Page 10 figure 2: It is impossible to see the genes, so I cannot get any information from this figure, do you mean only expression pattern?

Response: Yes, figure 2 shows expression pattern.

Page 10 line 213-215: I will be very careful to draw this conclusion. It is strange to say “successfully evolved” by only detecting the expression level of some certain genes that associated with certain functions. Besides, “successfully evolved” itself is not really appropriate, we would use “successfully survived”.

Response: We revised “successfully evolved” to “successfully survived by exploiting”.

Page 12 figure 3: Interesting results, maybe make the node labels bigger, it is hard to read.

Response: We made the figure node labels bigger and replaced the previous version.

Page 13 line 284: Citations needed!

Response: We have added the references. Please see line 296 and references 18 and 20.

Page 13 line 290: If you try to explain horizontal gene transfer, maybe not only use neighbor-joining for tree construction, and why you change to ML in HSP70?

Response: We performed both methods (NJ tree and ML tree) for LEA and HSP70 proteins, respectively. Both methods generated similar trees with slight differences. For

LEA proteins, the NJ tree clustered *A. avenae* LEA proteins well. Thus, we presented the NJ tree for the LEA proteins. For HSP70 proteins, we built the tree with HSP70 proteins from many species other than *A. avenae*. The ML tree with 1000 bootstrap replicates generated a better cluster, so we presented the ML tree for the HSP70 proteins.

Page 14 line 319: This section looks definitely not a discussion, it is more like a conclusion. I found a lot of discussion in your result section actually. I suggest you reorganize your result and discussion. Of course, with necessary citations in the text.

Response: Thank you for your suggestions. We revised the “Results” section title to the “Results and Discussion” section and “Discussion” section title to “Conclusion” section title.

Page 15 line 334: Are the nematodes cultured on the medium or other materials?

Response: The nematodes were cultured on *Rhizoctonia solani* reared on a substrate of autoclaved wheat. The information has been added to lines 346-347.

Page 15 line 336: So you mean the sucrose float is for all contaminants removal? Including fungus, right?

Response: Correct. To make sure there is no DNA contamination from the fungus *Rhizoctonia solani*, we downloaded the sixteen current available genomes of *Rhizoctonia solani* from the NCBI website and screened them against the assembled *A. avenae* genome using NCBI BLASTN. We found no fungus contamination in the *A. avenae* genome. The information has been added to the supplementary Note. Please see the below (lines 71-87 in the supplementary note) for details.

“To exclude DNA contaminations from *Rhizoctonia solani*, we screened the assembled *A. avenae* genome against sixteen current available genomes of *R. solani* that were downloaded from the NCBI database. The GenBank accession numbers of the sixteen *R. solani* genomes included GCA_016906535.1, GCA_017311305.1, GCA_015342405.1, GCA_015342435.1, GCA_015341985.1, GCA_015342415.1, GCA_000715385.1, GCA_000832345.2, GCA_000524645.1, GCA_000334115.1, GCA_001286725.1, GCA_900185085.1, GCA_000695385.1, GCA_001899475.2, GCA_000350255.1, GCA_003268435.1. BLASTN searches ($E_value \leq 1e-5$) identified ~ 180 average hits per *R. solani* genome (ranged from 23 hits to 264 hits). The Hits with aligned lengths above 1kb were identified as 18S ribosomal RNA homologs (percentages of identical matches: 75.6% ~ 77.8%). The hits with aligned lengths ranged from 100 bp to 600 bp were identified as parts of 18S, 28S ribosomal RNA homologs or highly conserved mRNA sequences that were also found in other species by manual BLASTN searches at the NCBI website. The majority of the hits showed less than 90 bp (percentages of identical matches < 100%). These data confirm

that there are no contaminations from *R. solani* DNA sequences in the assembled *A. avenae* genome.”

Page 16 line 348: Trimmed with what program? Trimmomatic?

Response: Yes, we used Trimmomatic. The software and reference were added. Please see line 362 in the main text.

Page 16 line 353: Citations for soap and newbler.

Response: The reference for soap is provided in the supplementary note. Newbler is provided by Roche 454 company and comes with the 454 sequencing instrument. There is no specific reference for Newbler software. We added its source in the supplementary note.

Page 16 line 356: I have not find the results from CEGMA.

Response: The CEGMA result is in lines 92-93.

Page 16 line 359: each RH for how long?

Response: For 24 hrs (Higa and Womersley, 1993).

Page 19 line 412-413: It will be better to add the website or FTP link here.

Response: We have added the website link for worm base website. Please see line 427.

Page 19 line 414: Need more information for phylogeny construction.

Response: Thank you for your suggestions. The information for LEA and HSP70 phylogeny constructions were summarized in the supplementary note. We added the following sentence: “Detailed methods for the phylogenetic tress of LEA and HSP70 proteins appear in **Supplementary Note.**” Please see lines 436-437.

Page 20 line 426: So largest 16 scaffolds instead of 23, why?

Response: It should be 23. We revised this.

Page 21 line 425: NCBI, right?

Response: Yes. According to your suggestion, we added NCBI in line 441.

Page 21 Reference: Some of them need formatted

Response: We have formatted the references according to the journal guidelines.

Reviewers' comments:

Reviewer #2 (Remarks to the Author):

I had previously remarked that the gene number of 43k is high and suggested to check the genome for polyploidy. The authors provide a kmer plot that seems to indicate diploidy. It would be good to also include a similar plot for the Illumina data, which is also the more correct datatype.

However, the authors - including in their letter - explicitly make the claim that *A. avenae* has a number of singletons far exceeding, for example, all species shown in Figure 1b. I think that they need to show greater effort to show that this claim is biological truth and not a fluke of their data analysis. Also as the number of 43k genes is still extremely high for any Metazoan species where a highly contiguous genome (i.e. with high N50) could be assembled. I see that the authors rely on SNAP for gene prediction, a software which is known to split multi exon genes into fragments. It should be noted that in such a case of splitting of genes, each fragment will still receive some RNA-Seq coverage and this line of evidence, used by the authors, cannot be relied on to prove a gene is correctly predicted.

There should also be a comparison to *Bursaphelenchus xylophilus* and potentially more clade IV species in regard to the gene numbers (e.g. 18,074 in *B. xylophilus*) and singletons.

In principle, I think it is not strictly necessary to re-annotate the genome for publication. However, the authors are basing substantial differential expression analysis on their gene set and also report on individual gene families. I fear that if there are many fragmented genes these analysis, which are at the core of this paper "Evolutionary Adaptation to Desiccation", the findings are not very robust indeed.

Minor point:

The genome must be made publicly available through Genbank and . This is not stated in the manuscript yet.

Reviewer #3 (Remarks to the Author):

My concerns have been taken care of nicely, so this manuscript now is acceptable to me.

Reviewers' comments:

Reviewer #2 (Remarks to the Author):

I had previously remarked that the gene number of 43k is high and suggested to check the genome for polyploidy. The authors provide a kmer plot that seems to indicate diploidy. It would be good to also include a similar plot for the Illumina data, which is also the more correct datatype.

However, the authors - including in their letter - explicitly make the claim that *A. avenae* has a number of singletons far exceeding, for example, all species shown in Figure 1b. I think that they need to show greater effort to show that this claim is biological truth and not a fluke of their data analysis. Also as the number of 43k genes is still extremely high for any Metazoan species where a highly contiguous genome (i.e. with high N50) could be assembled. I see that the authors rely on SNAP for gene prediction, a software which is known to split multi exon genes into fragments. It should be noted that in such a case of splitting of genes, each fragment will still receive some RNA-Seq coverage and this line of evidence, used by the authors, cannot be relied on to prove a gene is correctly predicted.

There should also be a comparison to *Bursaphelenchus xylophilus* and potentially more clade IV species in regard to the gene numbers (e.g. 18,074 in *B. xylophilus*) and singletons.

In principle, I think it is not strictly necessary to re-annotate the genome for publication. However, the authors are basing substantial differential expression analysis on their gene set and also report on individual gene families. I fear that if there are many fragmented genes these analysis, which are at the core of this paper "Evolutionary Adaptation to Desiccation", the findings are not very robust indeed.

Response: Thank you very much for your comments and suggestions. We fully agree that these questions are important to the fundamental basis of the main conclusion in this work. Please see the below for the explanations. Hope that we have answered these questions.

The kmer plot (Supplementary Figure 1) was carried out using all the sequencing reads for the genome assembly including Illumina data. The single peak of kmer distribution in Supplementary Figure 1 indicates that the reads are sequenced for a haploid genome. If a diploid heterozygous genome is sequenced, the kmer distribution of sequencing reads will show a bimodal distribution with two peaks. Please see [https://ucdavis-bioinformatics-training.github.io/2020-Genome Assembly Workshop/kmers/kmers](https://ucdavis-bioinformatics-training.github.io/2020-Genome%20Assembly%20Workshop/kmers/kmers). Although our flow cytometry analysis suggested that *A. avenae* has a diploid genome, the genome size estimation (255 Mb) based on flow cytometry analysis is for the haploid genome size. K-mer counting

estimation and the size of the assembled genome both support that a haploid genome is assembled. This may be because *A. avenae* has been reared for long term (decades) in the lab and *A. avenae* is able to reproduce asexually by parthenogenesis. Therefore, the two sets of chromosomes are highly homologous.

According to your suggestions, we have looked up genomes of species from clade IV and compared them to *A. avenae* using OrthoFinder (v2.3.9). We added this information to Supplementary Table 9, main text (lines 160-162) and Supplementary Note (lines 126-136). Serra et al. reported that the genome of *Steinernema carpocapsae* from clade IV comprises 84.5 Mb and encodes 30,957 genes (predicted by Augustus) (Serra et al. 2019, doi: <https://doi.org/10.1534/g3.119.400180>, 86.1 Mb and 31,937 genes in downloaded datasets, Supplementary Table 9). Moreover, the genome of *Meloidogyne incognita* (PRJEB8714, Blanc-Mathieu et al. 2017, doi: <https://doi.org/10.1371/journal.pgen.1006777>) comprises 183.5 Mb and encodes 43,718 genes. Compared to these numbers, it seems that 43k genes are not unacceptable.

For the concerns regarding fragmented genes, we analyzed the start and stop codons in 43k genes. Among them, 41,134 (95.2%) genes start with amino acid methionine, and 35,554 (82.3%) genes have stop codons, indicating that majority of the gene models are full genes.

For the concerns regarding singletons, both singleton (not member of any gene family) and genes in lineage-specific families in figure 1b are genes unique to the species among the compared genomes. Figure 1b shows that, besides *A. avenae*, genomes of other species also contain high percentages of gene numbers in singletons and genes in lineage-specific gene families, e.g. *M. hapla* and *T. spiralis* (together > 50%). Laing et al. reported that, when the genomes of nematodes from clade V were compared, *M. hapla* and *Pristionchus pacificus* showed these two types of genes with high percentages (together > 50%, Figure 1) (Laing et al. 2013, doi:10.1186/gb-2013-14-8-r88). In addition, although Laing et al. and we compared *B. xylophilus* to different sets of nematode genomes (Figure 1b), both works showed that the genome of *B. xylophilus* contains these two types of genes with similar percentages (nearly 50%).

According to your suggestions, we compared orthologous genes among clade IV species including *A. avenae*, *Steinernema carpocapsae*, *Meloidogyne hapla*, *Globodera pallida*, *Meloidogyne incognita* (PRJEB8714, population Morelos), *Meloidogyne incognita* (PRJNA340324, W1 strain), *Bursaphelenchus xylophilus* and *Panagrolaimus superbus* (Supplementary Table 9, listed below). Both *A. avenae* (47.7%) and *Steinernema carpocapsae* (50%) contain high percentages of species-specific genes (singletons and genes in lineage-specific gene families). *Panagrolaimus superbus* and *Bursaphelenchus xylophilus* contain 38% and 36.1% of species-specific genes, respectively. All these data suggest that the genomes of nematodes are highly variable in gene contents. This may be because these nematodes belong to different genera and reside in their species-specific ecosystems. Considering bacterial species from the same

genus, there are about hundreds of singletons (thousands of genes for each genome). Taken together, it is not unacceptable that a nematode genome has high percentage of species-specific genes when compared to other genera.

The revisions in the main text (lines 160-162), Supplementary Note (lines 126-136) and Supplementary Table 9 are listed below.

“Additional comparison of *A. avenae* and clade IV species showed that 20,615 (47.7%) proteins were unique to *A. avenae* (**Supplementary Note, Supplementary Table 9**).” (main text lines 160-162)

“To identify the orthologous genes and species-specific genes between *A. avenae* and clade IV species, we downloaded protein sequences of *Steinernema carpocapsae*⁸, *Meloidogyne hapla*⁹, *Globodera pallida*¹⁰, *Meloidogyne incognita* (population Morelos)¹¹, *Meloidogyne incognita* (W1 strain)¹², *Bursaphelenchus xylophilus*¹³ and *Panagrolaimus superbus*¹⁴ from WormBase ParaSite website (<https://parasite.wormbase.org/index.html>, version: WBPS15 (WS276)). OrthoFinder v2.3.9^{15,16} employing MCL clustering algorithm was carried out for ortholog inference. The comparison results were listed in **Supplementary Table 9**. Both *A. avenae* (47.7%) and *S. carpocapsae* (50%) showed high percentages of species-specific genes (singletons and genes in lineage-specific gene families), indicating that the genomes of nematodes, even from the same clade, are highly variable in gene contents.” (Supplementary Note lines 126-136)

Supplementary Table 9 Comparison of orthologous genes between *A. avenae* and clade IV species

	A. avenae	Steinernema carpocapsae	Meloidogyne hapla	Globodera pallida	Meloidogyne incognita (a)	Meloidogyne incognita (b)	Bursaphelenchus xylophilus	Panagrolaimus superbus
Total size of assembled genome (Mb)	264.8	86.1	53	123.6	183.5	122	74.6	76.7
Gene models	43192	31937	14419	16403	43718	21830	17704	19663
Gene density (genes per Mb)	163	371	272	133	238	179	237	256
proteins in 8 spp. shared families (percentage)	8532 (19.8%)	6882 (21.5%)	4658 (32.3%)	5533 (33.7%)	13840 (31.7%)	7867 (36.0%)	4975 (28.1%)	5730 (29.1%)
proteins in other multi-gene families (percentage)	14045 (32.5%)	9081 (28.4%)	8152 (56.5%)	5202 (31.7%)	23564 (53.9%)	11912 (54.6%)	6328 (35.7%)	6469 (32.9%)
proteins in lineage-specific gene families (percentage)	17637 (40.8%)	8724 (27.3%)	743 (5.2%)	4166 (25.4%)	3756 (8.6%)	805 (3.7%)	3331 (18.8%)	3890 (19.8%)
singleton proteins (percentage)	2978 (6.9%)	7250 (22.7%)	866 (6.0%)	1502 (9.2%)	2558 (5.9%)	1246 (5.7%)	3070 (17.3%)	3574 (18.2%)

(a) PRJEB8714, population Morelos

(b) PRJNA340324, W1 strain

Minor point:

The genome must be made publicly available through Genbank and . This is not stated in the manuscript yet.

Response: Thank you very much for your suggestions. Please see lines 469-472 for the revised data and materials availability: “The sequencing data are deposited in the Sequence Read Archive database (PRJNA236621, PRJNA236622). The genome assembly with gene annotations is deposited under PRJNA236621. All data are available in the manuscript or the supplementary materials. Materials are available upon request.” We will release all the data including the genome assembly and annotated genes (under BioProject accession number PRJNA236621) once the manuscript is accepted for publication.

Reviewers' comments:

Reviewer #2 (Remarks to the Author):

I had previously remarked that the gene number of 43k is high and suggested to check the genome for polyploidy. The authors provide a kmer plot that seems to indicate diploidy. It would be good to also include a similar plot for the Illumina data, which is also the more correct datatype.

However, the authors - including in their letter - explicitly make the claim that *A. avenae* has a number of singletons far exceeding, for example, all species shown in Figure 1b. I think that they need to show greater effort to show that this claim is biological truth and not a fluke of their data analysis. Also as the number of 43k genes is still extremely high for any Metazoan species where a highly contiguous genome (i.e. with high N50) could be assembled. I see that the authors rely on SNAP for gene prediction, a software which is known to split multi exon genes into fragments. It should be noted that in such a case of splitting of genes, each fragment will still receive some RNA-Seq coverage and this line of evidence, used by the authors, cannot be relied on to prove a gene is correctly predicted.

There should also be a comparison to *Bursaphelenchus xylophilus* and potentially more clade IV species in regard to the gene numbers (e.g. 18,074 in *B. xylophilus*) and singletons.

In principle, I think it is not strictly necessary to re-annotate the genome for publication. However, the authors are basing substantial differential expression analysis on their gene set and also report on individual gene families. I fear that if there are many fragmented genes these analysis, which are at the core of this paper "Evolutionary Adaptation to Desiccation", the findings are not very robust indeed.

Response: Thank you very much for your comments and suggestions. We fully agree that these questions are important to the fundamental basis of the main conclusion in this work. Please see the below for the explanations. Hope that we have answered these questions.

The kmer plot (Supplementary Figure 1) was carried out using all the sequencing reads for the genome assembly including Illumina data. The single peak of kmer distribution in Supplementary Figure 1 indicates that the reads are sequenced for a haploid genome. If a diploid heterozygous genome is sequenced, the kmer distribution of sequencing reads will show a bimodal distribution with two peaks. Please see [https://ucdavis-bioinformatics-training.github.io/2020-Genome Assembly Workshop/kmers/kmers](https://ucdavis-bioinformatics-training.github.io/2020-Genome%20Assembly%20Workshop/kmers/kmers). Although our flow cytometry analysis suggested that *A. avenae* has a diploid genome, the genome size estimation (255 Mb) based on flow cytometry analysis is for the haploid genome size. K-mer counting

estimation and the size of the assembled genome both support that a haploid genome is assembled. This may be because *A. avenae* has been reared for long term (decades) in the lab and *A. avenae* is able to reproduce asexually by parthenogenesis. Therefore, the two sets of chromosomes are highly homologous.

According to your suggestions, we have looked up genomes of species from clade IV and compared them to *A. avenae* using OrthoFinder (v2.3.9). We added this information to Supplementary Table 9, main text (lines 160-162) and Supplementary Note (lines 126-136). Serra et al. reported that the genome of *Steinernema carpocapsae* from clade IV comprises 84.5 Mb and encodes 30,957 genes (predicted by Augustus) (Serra et al. 2019, doi: <https://doi.org/10.1534/g3.119.400180>, 86.1 Mb and 31,937 genes in downloaded datasets, Supplementary Table 9). Moreover, the genome of *Meloidogyne incognita* (PRJEB8714, Blanc-Mathieu et al. 2017, doi: <https://doi.org/10.1371/journal.pgen.1006777>) comprises 183.5 Mb and encodes 43,718 genes. Compared to these numbers, it seems that 43k genes are not unacceptable.

For the concerns regarding fragmented genes, we analyzed the start and stop codons in 43k genes. Among them, 41,134 (95.2%) genes start with amino acid methionine, and 35,554 (82.3%) genes have stop codons, indicating that majority of the gene models are full genes.

For the concerns regarding singletons, both singleton (not member of any gene family) and genes in lineage-specific families in figure 1b are genes unique to the species among the compared genomes. Figure 1b shows that, besides *A. avenae*, genomes of other species also contain high percentages of gene numbers in singletons and genes in lineage-specific gene families, e.g. *M. hapla* and *T. spiralis* (together > 50%). Laing et al. reported that, when the genomes of nematodes from clade V were compared, *M. hapla* and *Pristionchus pacificus* showed these two types of genes with high percentages (together > 50%, Figure 1) (Laing et al. 2013, doi:10.1186/gb-2013-14-8-r88). In addition, although Laing et al. and we compared *B. xylophilus* to different sets of nematode genomes (Figure 1b), both works showed that the genome of *B. xylophilus* contains these two types of genes with similar percentages (nearly 50%).

According to your suggestions, we compared orthologous genes among clade IV species including *A. avenae*, *Steinernema carpocapsae*, *Meloidogyne hapla*, *Globodera pallida*, *Meloidogyne incognita* (PRJEB8714, population Morelos), *Meloidogyne incognita* (PRJNA340324, W1 strain), *Bursaphelenchus xylophilus* and *Panagrolaimus superbus* (Supplementary Table 9, listed below). Both *A. avenae* (47.7%) and *Steinernema carpocapsae* (50%) contain high percentages of species-specific genes (singletons and genes in lineage-specific gene families). *Panagrolaimus superbus* and *Bursaphelenchus xylophilus* contain 38% and 36.1% of species-specific genes, respectively. All these data suggest that the genomes of nematodes are highly variable in gene contents. This may be because these nematodes belong to different genera and reside in their species-specific ecosystems. Considering bacterial species from the same

genus, there are about hundreds of singletons (thousands of genes for each genome). Taken together, it is not unacceptable that a nematode genome has high percentage of species-specific genes when compared to other genera.

The revisions in the main text (lines 160-162), Supplementary Note (lines 126-136) and Supplementary Table 9 are listed below.

“Additional comparison of *A. avenae* and clade IV species showed that 20,615 (47.7%) proteins were unique to *A. avenae* (**Supplementary Note, Supplementary Table 9**).” (main text lines 160-162)

“To identify the orthologous genes and species-specific genes between *A. avenae* and clade IV species, we downloaded protein sequences of *Steinernema carpocapsae*⁸, *Meloidogyne hapla*⁹, *Globodera pallida*¹⁰, *Meloidogyne incognita* (population Morelos)¹¹, *Meloidogyne incognita* (W1 strain)¹², *Bursaphelenchus xylophilus*¹³ and *Panagrolaimus superbus*¹⁴ from WormBase ParaSite website (<https://parasite.wormbase.org/index.html>, version: WBPS15 (WS276)). OrthoFinder v2.3.9^{15,16} employing MCL clustering algorithm was carried out for ortholog inference. The comparison results were listed in **Supplementary Table 9**. Both *A. avenae* (47.7%) and *S. carpocapsae* (50%) showed high percentages of species-specific genes (singletons and genes in lineage-specific gene families), indicating that the genomes of nematodes, even from the same clade, are highly variable in gene contents.” (Supplementary Note lines 126-136)

Supplementary Table 9 Comparison of orthologous genes between *A. avenae* and clade IV species

	A. avenae	Steinernema carpocapsae	Meloidogyne hapla	Globodera pallida	Meloidogyne incognita (a)	Meloidogyne incognita (b)	Bursaphelenchus xylophilus	Panagrolaimus superbus
Total size of assembled genome (Mb)	264.8	86.1	53	123.6	183.5	122	74.6	76.7
Gene models	43192	31937	14419	16403	43718	21830	17704	19663
Gene density (genes per Mb)	163	371	272	133	238	179	237	256
proteins in 8 spp. shared families (percentage)	8532 (19.8%)	6882 (21.5%)	4658 (32.3%)	5533 (33.7%)	13840 (31.7%)	7867 (36.0%)	4975 (28.1%)	5730 (29.1%)
proteins in other multi-gene families (percentage)	14045 (32.5%)	9081 (28.4%)	8152 (56.5%)	5202 (31.7%)	23564 (53.9%)	11912 (54.6%)	6328 (35.7%)	6469 (32.9%)
proteins in lineage-specific gene families (percentage)	17637 (40.8%)	8724 (27.3%)	743 (5.2%)	4166 (25.4%)	3756 (8.6%)	805 (3.7%)	3331 (18.8%)	3890 (19.8%)
singleton proteins (percentage)	2978 (6.9%)	7250 (22.7%)	866 (6.0%)	1502 (9.2%)	2558 (5.9%)	1246 (5.7%)	3070 (17.3%)	3574 (18.2%)

(a) PRJEB8714, population Morelos

(b) PRJNA340324, W1 strain

Minor point:

The genome must be made publicly available through Genbank and . This is not stated in the manuscript yet.

Response: Thank you very much for your suggestions. Please see lines 469-472 for the revised data and materials availability: “The sequencing data are deposited in the Sequence Read Archive database (PRJNA236621, PRJNA236622). The genome assembly with gene annotations is deposited under PRJNA236621. All data are available in the manuscript or the supplementary materials. Materials are available upon request.” We will release all the data including the genome assembly and annotated genes (under BioProject accession number PRJNA236621) once the manuscript is accepted for publication.

REVIEWERS' COMMENTS:

Reviewer #3 (Remarks to the Author):

According to the additional analysis and explanations the authors provided, I think the concerns from reviewer2 are solved.